# A causal relationship between face-patch activity and face-detection behavior

Srivatsun Sadagopan[1], Wilbert Zarco[2], Winrich A Freiwald[2]*

[1]Departments of Otolaryngology and Bioengineering, University of Pittsburgh, Pittsburgh, United States; [2]Laboratory of Neural Systems, The Rockefeller University, New York, United States

**Abstract** The primate brain contains distinct areas densely populated by face-selective neurons. One of these, face-patch ML, contains neurons selective for contrast relationships between face parts. Such contrast-relationships can serve as powerful heuristics for face detection. However, it is unknown whether neurons with such selectivity actually support face-detection behavior. Here, we devised a naturalistic face-detection task and combined it with fMRI-guided pharmacological inactivation of ML to test whether ML is of critical importance for real-world face detection. We found that inactivation of ML impairs face detection. The effect was anatomically specific, as inactivation of areas outside ML did not affect face detection, and it was categorically specific, as inactivation of ML impaired face detection while sparing body and object detection. These results establish that ML function is crucial for detection of faces in natural scenes, performing a critical first step on which other face processing operations can build.

*For correspondence: wfreiwald@ mail.rockefeller.edu

**Competing interests:** The authors declare that no competing interests exist.

## Introduction

For primates, faces are among the most behaviorally salient and socially relevant visual stimuli (*Leopold and Rhodes, 2010*). Reflecting this importance , the primate brain contains multiple interconnected cortical areas specialized for face processing, the face-patch network (*Tsao et al., 2006, 2008*). The middle-lateral face patch (ML), one of the earliest nodes in the face processing hierarchy, consists of neurons highly selective for faces (*Tsao et al., 2006*). Half of ML neurons exhibit sensitivity for coarse contrast relationships between face regions such as vertical adjacency of a bright forehead region above a darker eye region (*Ohayon et al., 2012*). Such qualitative contrast signatures for faces have been shown to provide a computationally powerful means for face detection (*Sinha, 2002*; *Viola and Jones, 2004*; *Gilad et al., 2009*), the most fundamental computation for all subsequent face-processing (*Tsao and Livingstone, 2008*). The hierarchically early location of face patch ML and the functional specificity of ML neurons led us to hypothesize that ML neurons might be critical for detection of faces, but not objects, in natural scenes. Testing this hypothesis required development of a naturalistic and demanding face and object detection task, and monitoring of real-world face-detection performance during causal manipulation of face patch ML.

## Results and discussion

We first trained two macaques on a touchscreen-based object detection task. From a visual scene displayed on the touchscreen, subjects were required to select an object belonging to one of three target categories – human faces, macaque bodies (socially relevant control category represented near face patches; *Popivanov et al., 2012*), or shoes (newly learned object category that, like faces, exhibits high within-class similarity). The visual scene was composed of 24 objects, at one of ten visibility levels. To discourage the use of low-level features for solving the task, the face targets were

randomly resized, rotated in-plane, and blended into a cluttered background at random locations (*Figure 1A*; see Materials and methods). Face patch ML was localized by functional magnetic resonance imaging (fMRI) using a standard face localizer. To inactivate ML, we stereotaxically targeted its fMRI-identified coordinates and infused 5 µL (8 µL in one experiment) of muscimol (5 µg/µL), a $GABA_A$ agonist. We co-infused the MR contrast agent gadolinium (Gd), the cortical spread of which closely tracks the diffusion extent of muscimol (*Heiss et al., 2010*; *Wilke et al., 2012*), as a surrogate tracer so that the extent of muscimol spread could be detected in an anatomical MR scan (*Figure 1B*). Overlap of muscimol diffusion with the functionally defined extent of ML was determined post-hoc by aligning the post-injection anatomical volume and previous functional scan data (see Materials and methods; *Figure 1B–D*).

The goal of the injections was to inactivate as much of ML as possible, with as little inactivation of outside regions as possible – with the converse goal for the off-target injections. To determine how well these goals were met, we computed the percentage overlap of injections with ML and the

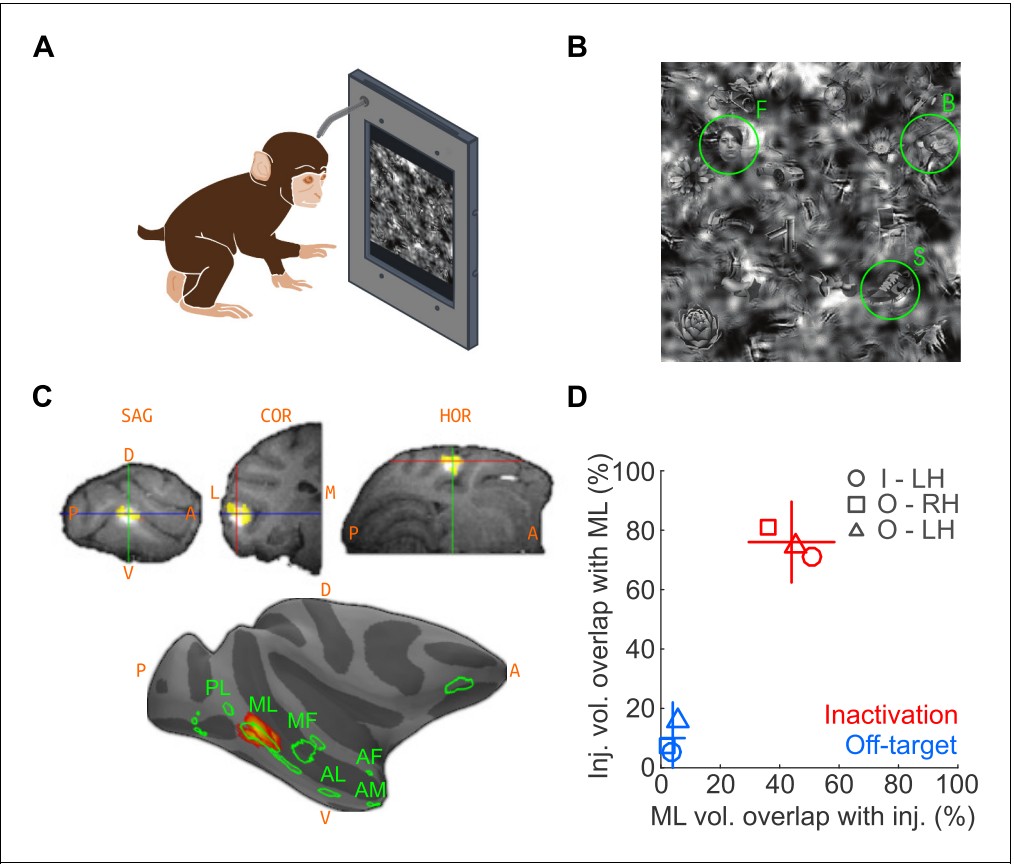

**Figure 1.** Naturalistic face-detection task and targeted inactivation of face-patch ML. (**A**) Subjects performed a face-detection task on a touchscreen device that was mounted to their home cages. Subjects were unconstrained, initiated trials at will, and scanned the array freely. (**B**) Upon initiating a trial, an array of 24 objects appeared on the screen, one of which was the target belonging to one of three categories – faces, bodies, or shoes (green circles). (**C**) Neural activity in face-patch ML (yellow regions) was silenced by targeted microinjections of muscimol. (*Top*) The extent of diffusion and targeting accuracy was evaluated by coinjecting gadolinium, which appeared as a bright volume in post-injection anatomical scans. Labels: SAG - sagittal slice, HOR - horizontal slice, COR - coronal slice, A - anterior, P - posterior, D - dorsal, V – ventral. (*Bottom*) The locations of all four targeted muscimol microinjections (heatmap) from one subject's left hemisphere relative to face-patch locations (green outlines, face area labels as in *Tsao et al., 2008*). (**D**) Volumetric overlap between the inactivation injections and ML (red; n = 11) and off-target injections and ML (blue; n = 10). Lines correspond to ±1 s.e.m., and the intersection point is the mean along each axis. Symbols correspond to the mean values from individual hemispheres (circle – Subject 'I', LH; square – Subject 'O', RH; triangle – Subject 'O', LH).

fraction of ML volume overlapping with the injection (*Figure 1D*). First, we determined the average volume of ML exceeding a threshold of t > 20 as ~58 mm$^3$, corresponding to a ~5 mm-diameter sphere, consistent with published neurophysiological data (*Aparicio et al., 2016*). The average volume of the injection at a threshold of 50% peak Gd brightness was ~37 mm$^3$, corresponding to a ~4 mm-diameter sphere. At the chosen thresholds, the theoretical maximum of volumetric ML coverage by the injection, assuming concentric spheres for both volumes, is 63.8%. We achieved 44% coverage on average. Of the total injection volume, 76% was contained within the ML volume (*Figure 1D*). The high overlap of target injections with ML suggests that we silenced a large fraction of highly face-selective neurons, primarily found in the center of the face patches (*Aparicio et al., 2016*). The corresponding numbers for off-target injections were 4% ML coverage with 10% of the injection contained within ML. It is thus unlikely that off-target injections, although overlapping slightly with ML, silenced many face-selective neurons. The subject was transferred to its home cage to perform the detection tasks between ~1 and 3 hr after muscimol injection and subsequent anatomical scan, well within the time period of maximal muscimol effects (~0.5 hr to ~4 hr) (*Hikosaka and Wurtz, 1985*; *Lomber, 1999*; *Dias and Segraves, 1999*; *Sommer and Wurtz, 2004*). In a typical experimental sequence, we gathered baseline performance data on Day 1, inactivation data on Day 2, and recovery data on Day 3 or 4.

We first compared average face-detection performance across all 11 inactivation sessions (two monkeys, three hemispheres) and their corresponding baseline sessions. Face-detection performance was reduced by ~11% during ML inactivation (pooled over all visibilities and sessions, p<0.001, paired t-test after logit transform). Separated by visibility level, these performance decrements were significant for 9 out of 10 visibility levels at p<0.05 (FDR-corrected paired t-tests after logit transform; *Figure 2A*). A comparison of single session performances during baseline and inactivation showed this performance reduction to be uniform across visibility levels (*Figure 2B*): it was well-described by a line (R$^2$ = 0.74; p<0.001) parallel to the diagonal (slope = 0.96) with a significant negative offset (p<0.001), providing a second, visibility-independent, single-session level estimate of the performance deficit of ~15%. In *Figure 2C and D*, we plot the same quantities as above separated into each hemisphere and animal to demonstrate that the reduction in performance was consistent between subjects and inactivated hemispheres. At the single-hemisphere level, we observed significant differences mainly at intermediate visibility levels (p<0.05; FDR-corrected paired t-test after logit transform). Thus, inactivation of 1 out of 12 temporal lobe face patches (*Tsao et al., 2008*) caused a significant and sizable reduction in face-detection performance.

To determine which behavioral parameters were affected by inactivation, psychometric functions (*Equation 1* in Materials and methods; *Wichmann and Hill, 2001*) with three parameters – shift along the abscissa ($\alpha$), slope ($\beta$), and lapse rate ($\lambda$, the difference between maximum and perfect performance) - were then fit to each session's performance data (R$^2$ > 0.9 for face detection in all sessions). Shift ($\alpha$) and lapse rate ($\lambda$) were significantly higher in the inactivated conditions (p<0.001 and p = 0.016 respectively; paired t-test) than during baseline (red disks in *Figure 2E*; *Figure 2F*). In other words, during inactivation, the psychometric function was shifted rightwards and asymptotically lowered. The slope of the function ($\beta$) remained unaltered by inactivation. As a result, the stimulus visibility required for 50% performance (the average face-detection threshold) increased from 0.37 ± 0.02 at baseline to 0.45 ± 0.03 during the inactivation conditions.

We next tested whether the behavioral impairment following inactivation was specific to faces. *Figure 3A* shows the cumulative (summed across sessions) performance curves for face (left), body (middle), and shoe (right) detection. ML inactivation caused a shift of the psychometric function for face detection, but neither body nor shoe detection were significantly impaired. Effects were specific to the injection both in time and space. In recovery sessions when the effect of muscimol was expected to have subsided, behavioral performance returned to baseline levels across all target categories (*Figure 3B*; green), and fit parameters for performance in the recovery condition were not different from baseline for faces (*Figure 2E*; *Figure 2F*; green) or the other target categories (not shown). The transience of this effect demonstrates that muscimol was a causative agent. We also injected muscimol into cortex or white matter near ML (off-target injection, see Materials and methods). The off-target injection volumes barely overlapped with ML (9.6%; *Figure 1D*) or with any other face patch, and consequently, we observed only a weak and non-significant reduction in detection performance (*Figure 2E*; *Figure 2F*; *Figure 3B*; blue).

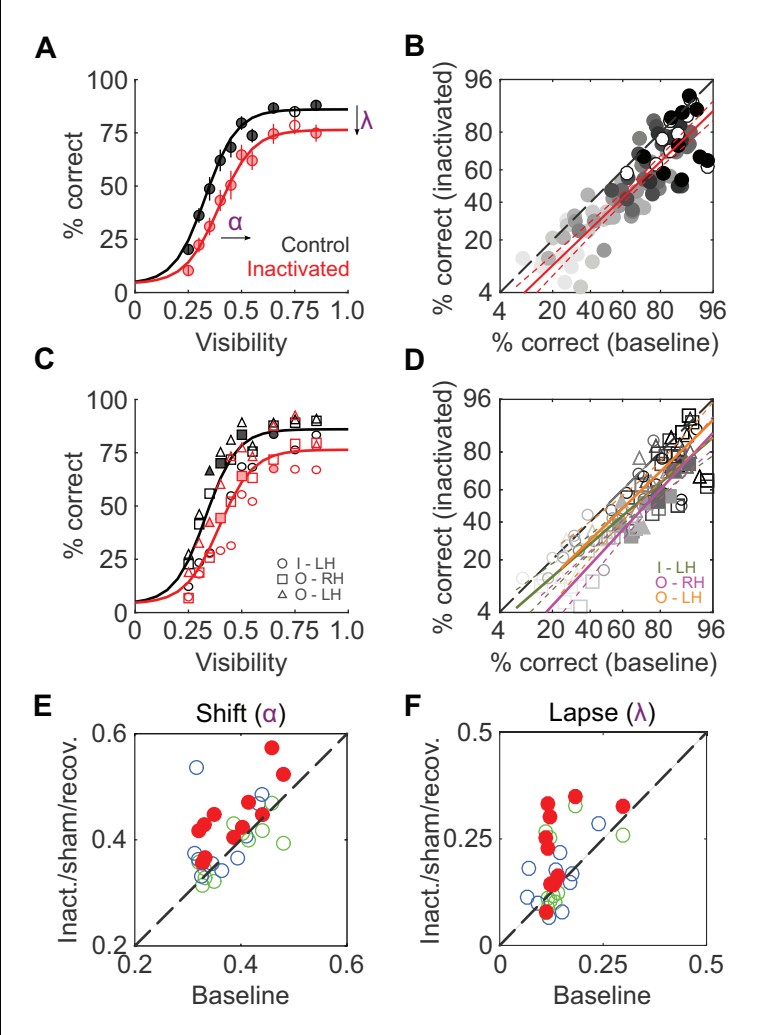

**Figure 2.** ML inactivation causes face-detection deficits. (**A**) Face-detection performance as a function of stimulus visibility under baseline (black) and ML-inactivated (red) conditions. Circles correspond to average performance, error bars to ±1 s.e.m. Lines are psychometric function fits to all performance data. Filled circles: p<0.05, FDR-corrected paired t-tests after logit transform. (**B**) Performance in the inactivated condition plotted as a function of performance in the corresponding baseline condition, sorted by stimulus visibility (grayscale luminance). Filled circles are the significant points from (**A**). These data were well-fit by a line (red; $R^2$ = 0.74; p<0.001) with a slope close to 1, and a significantly shifted intercept (p<0.001) compared withthe diagonal. Dashed red lines represent the 95% confidence intervals. (**C and D**) The data in (**A**) and (**B**), but separated into individual inactivated hemispheres. Filled symbols denote statistically significant differences between the control and inactivated conditions (p<0.05, FDR-corrected paired t-test after logit transform). Symbols are as in *Figure 1*. (**E and F**) Face-detection performance as quantified by shift α (**E**) and lapse λ (**F**) during three conditions (ML-inactivated (red), recovery (green) and off-target (blue)) versus baseline condition. Shift and lapse were derived from psychometric function fits to face-detection data, and were significantly altered from baseline during inactivation (filled red circles; p<0.001 for shift and p<0.05 for lapse; paired t-test). Note that the axes in (**B**) and (**D**) are logit scaled.

We then compared the grand average performance deficits (across all stimulus visibilities and sessions) and threshold shifts (across all sessions) for all category/condition combinations (*Figure 3C and D*). We used an ANCOVA with each target category and experimental condition as an independent group (seven groups) and muscimol exposure session number as a covariate, with all fits constrained to have equal slopes on logit-transformed data to evaluate statistical significance. Only the face-inactivated condition showed significant performance deficits ($F_{(6, 752)}$=17.89; p<0.0001) and

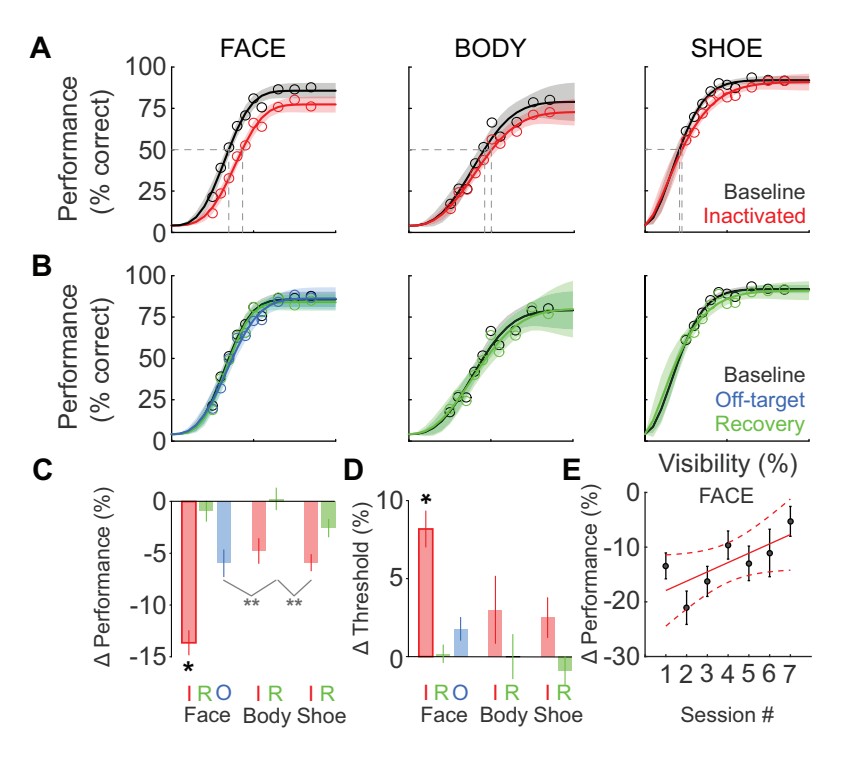

**Figure 3.** Deficits after ML inactivation are reversible and largely categorically specific to faces. (**A**) Behavioral performance in the baseline (black) and ML-inactivation (red) conditions for face-, body-, and shoe-detection as a function of stimulus visibility, showing the categorical specificity of the effect to faces. Data are summed across all 11 inactivation sessions. Lines are fits to the psychometric function in *Equation 1*. Shading shows the 95% confidence intervals. Gray dashed lines indicate threshold stimulus visibilities for each stimulus category. (**B**) Behavioral performance measured the day after ML inactivation (recovery; green) was not different from the baseline condition (black). Behavioral performance when muscimol was infused into the surrounding cortical areas, but not into ML (off-target; blue), was not different from the baseline condition, showing the anatomical specificity of the effect. (**C**) The performance deficit, averaged across all visibilities and sessions, for all tested category/condition combinations (colors as earlier). Face/inactivated performance deficit was significantly different for all pairwise comparisons (ANCOVA accounting for effect of repeated muscimol exposure on logit-transformed data; black asterisk: p<0.05, Tukey's test). An ANOVA not accounting for repeated exposure resulted in two cross-category comparisons reaching significance (gray asterisks: p<0.01, Tukey's test). (**D**) The change in detection threshold for all tested category/condition combinations. The face/inactivated threshold was significantly shifted to higher stimulus visibilities for all pairwise comparisons (ANOVA; black asterisk, p<0.05, Tukey's test). Labels 'I', 'R', and 'O' refer to the inactivated, recovery, and off-target conditions. (**E**) The effect of repeated muscimol exposure on the performance deficit caused by muscimol. Dots correspond to mean effect pooled over visibilities, error bars are s.e.m. Red line corresponds to linear fit, dashed lines correspond to 95% confidence intervals.

threshold shifts (ANOVA; F(6, 65)=5.74; p<0.0001) that were pairwise significantly different from all of the other category/condition combinations including the body- and shoe-inactivated conditions (*p<0.05; Tukey's test). The mean face-detection performance deficit was not significantly correlated with the ML-injection overlap (Spearman's Rho $r_s$ = 0.42; p=0.1). An ANCOVA analysis with session number as independent groups and ML-injection overlap as a covariate did not reveal a significant effect of ML-injection overlap on the observed face-detection deficit (F(1,102) = 1.35; p=0.25). These data suggest that although there are differences in the extent of ML-injection overlap between sessions, as the central portion of ML was targeted by the injection, performance deficits were largely unaffected by these small differences in injection volume and overlap. Using an ANOVA (on logit-transformed data) that did not account for the effect of repeated muscimol exposure, we observed two additional pairwise comparisons that showed cross-category significance (light gray lines in *Figure 3C*; **p<0.01; Tukey's test), but which did not reach significance for the appropriate within-

category comparisons. Thus ML inactivation is largely category-selective for faces, and anatomically specific for ML compared with its immediate surroundings.

Although ML inactivation did not cause statistically significant performance deficits or threshold shifts for the other object categories, it is possible that the data might contain a non-specific component, as indicated by the non-zero values of performance deficits and threshold shifts for body parts and shoes (*Figure 3C and D*). To gain more statistical power for testing for a non-specific effect, we repeated the ANCOVA analysis that yielded the statistically significant performance deficit in *Figure 3C*, but with the 'body' and 'shoe' categories grouped into a single 'non-face' category. This procedure also failed to yield statistically significant differences between 'non-face' control and inactivated conditions. Similarly, grouping the 'body' and 'shoe' categories in the ANOVA analysis of threshold failed to yield a significant difference in threshold. If a weak non-specific effect truly existed, we could not attribute this to the motivation level of the animal post injection and MRI scan, as there were no significant differences between the total number of trials the animal initiated on control or inactivation days (paired t-test, p=0.47). Alternately, although 76% of the injection volume was contained within ML, it is possible that some muscimol diffused out of ML and into neighboring body patches (*Popivanov et al., 2012*) in some cases, causing a smaller, non-significant effect on that object category.

Our behavioral paradigm was designed to emphasize natural face detection, thus allowing free viewing. The paradigm therefore encouraged the use of behavioral strategies that made maximal use of face-processing resources in the unaffected hemisphere. Compatible with the idea that such strategies might have been learned and used, we found muscimol-induced performance deficits to decrease with repeated muscimol exposure. This accounted for some of the variability seen in *Figure 2B*, and could be quantified using the ANCOVA analysis described above. We observed a highly significant effect of session number on the performance deficit magnitude (F(1,752)=15.33; p<0.0001) with a positive slope of 0.01 ± 0.002, that is, the performance deficit caused by muscimol decreased by ~1% each session (relative to baseline performance level; *Figure 3E*). It is also possible that over the course of the inactivation experiments, animals gradually learned to use stimulus features represented outside of ML such as low-level features of faces, despite our efforts to prevent this through scaling and in-plane rotation of targets and varying the texture backgrounds. When we performed the ANCOVA with subject identity as a covariate, we did not observe a significant effect, demonstrating that the gradual reduction of the effect of muscimol and the variability in the data (*Figure 2B*) did not arise from pooling data from the two subjects.

To illustrate the putative perceptual impact of ML inactivation on face detection, we derived equivalent inactivated visibility values from the psychometric curves (*Figure 4A*; similar to *Carrasco et al., 2004*). At a high visibility of 0.85 (gray dashed line, marked '1' in *Figure 4A*), performance in the inactivated condition was ~75% correct, and the same performance level was achieved in the baseline condition at a visibility of 0.49 (red dashed line, marked '1' in *Figure 4A*). Therefore, when stimulus visibility was 0.85 in the inactivated condition, the subject was behaving *as if* the visibility was 0.49, and so forth (*Figure 4B*). As expected from the increased detection threshold, equivalent inactivated visibilities for faces were significantly lower than baseline visibilities (p<0.01; paired t-test). In *Figure 4C*, we plot example face stimuli for two visibility pairs (filled circles in *Figure 4B*). Thus, the induced shift in the psychometric curve is potentially reflected as a perceptually profound reduction of equivalent face visibility.

The free-viewing nature of the task exploits natural visual search behavior to demonstrate the causal dependence of this behavior on face-patch activity. This design is similar to one used in many human visual search and object recognition studies, but, because reliable methods for eye-tracking in freely moving primates are not available, we did not track the eye position of subjects as they performed the task. Recent evidence suggests that object recognition performance in monkeys does not depend on response modality – performance measured using saccades with eye-tracking, or touch with no gaze tracking, was similar to human object recognition performance measured without eye-tracking (*Rajalingham et al., 2015*). Therefore, we do not expect the lack of eye-tracking to affect the main results of our study.

Lack of eye-tracking, however, introduces two main sources of variability into our data. First, we cannot clearly assess the spatial effects of unilateral inactivation. Because we inactivated ML unilaterally in our experiments, we attempted to address whether performance deficits were also lateralized by analyzing recognition performance when the target image was located in three vertically oriented

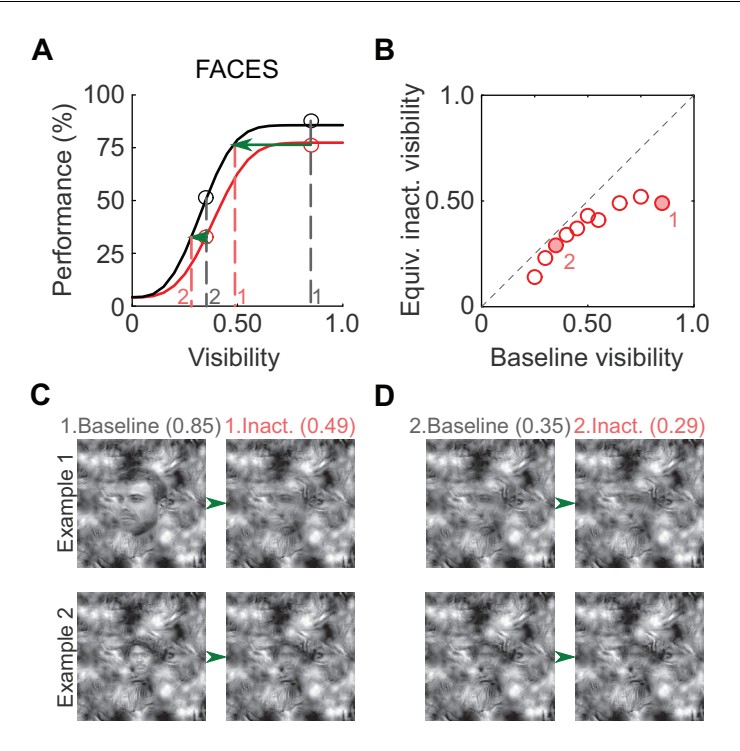

**Figure 4.** Perceptual effects of ML inactivation. (**A**) Performance under baseline (black) and inactivated (red) conditions for face detection reproduced from *Figure 2A* (summed data). Dotted lines outline how equivalent inactivated visibility was calculated. For example, the ML-inactivated performance at a stimulus visibility of 0.85 (gray dashed line and gray '1') was the same as baseline performance when stimulus visibility was 0.49 (red dashed line and red '1'), that is, when ML was inactivated, the subject behaved as if it was viewing the stimulus at a decreased visibility. (**B**) As expected from the shifted psychometric curves, equivalent inactivated visibility was always lower than baseline visibility at each of the visibility levels tested (p=0.002; paired t-test). Filled circles marked '1' and '2' correspond to the visibility levels marked in **A** and **C**. (**C and D**) Example face stimuli plotted at baseline (0.85, marked '1', and 0.35, marked '2') and equivalent inactivated visibility levels corresponding to these baseline levels.

bands – ipsilateral to injection site, contralateral, and in the center of the screen. This analysis reduced our overall statistical power and did not yield significant results. Previous experiments in head-fixed preparations have demonstrated that local inactivation of smaller cortical regions can cause face-processing deficits specific to the visual field they represent (e.g., *Afraz et al., 2015*). As face-patch ML shows a foveal bias (*Janssens et al., 2014*), and because the center of ML contains neurons with the strongest face selectivity (*Aparicio et al., 2016*), by targeting our injections to the center of ML, we may have interfered with face detection in a large foveal area that would force the subject to use alternate and slower face-detection strategies. Thus performance deficits may have been higher at target locations that correspond to retinotopic locations of the injection, and effect sizes reported here should be taken as lower bound estimates.

Second, the lack of eye-tracking limits our ability to quantify behavioral strategies that may have been used by the subject to solve the task. In particular, it adds variability to the psychometric curve and affects our estimate of the 'chance level' of the task because the subject may have looked away from the screen after initiating a trial. We designed our task to minimize this variability by: (1) requiring the animals to voluntarily initiate trials; (2) requiring subjects to direct their gaze to the center of the screen and perform a deliberate and accurate movement to initiate the trial; (3) requiring subjects to pay attention to the color of the central cue to determine target block identity; (4) requiring subjects to complete the task in a short time span; and (5) randomizing visibility levels within blocks and interleaving block types to prevent systematic biases. Through post-hoc analysis we determined

that: (1) the animal performed similar numbers of trials on control and inactivation days suggesting no change in motivation level; (2) animals performed at a high rate on high-visibility trials on all days, suggesting no distraction at the time scale of trials. The experimental configuration ensured that the screen covered a large portion of the animal's visual field, minimizing space available for distractions. Therefore, we believe that we have minimized the effects of extraneous sources on behavior. Additional sources of variability may remain, however, such as variations in eccentricity of the target stimulus as the subject freely views the display.

Direct cortical stimulation and suppression have been used successfully to alter face perception in humans and macaques (*Allison et al., 1994*; *Parvizi et al., 2012*; *Chong et al., 2013*; *Jonas et al., 2014*; *Afraz et al., 2006*; *Roy et al., 2014*; *Afraz et al., 2015*). We provide four critical advances over these past experiments: (1) We directly visualized and limited the spread of muscimol to the functionally defined coordinates of an fMRI-identified face area. Comparing on- and off-target injections, we could thus establish anatomical specificity of inactivation. (2) We used several interleaved object categories as behavioral targets. We could thus establish categorical specificity of ML inactivation. (3) We focused on face detection, the most fundamental and robust component of face processing (*Sinha, 2002*; *Viola and Jones, 2004*; *Tsao and Livingstone, 2008*; *Gilad et al., 2009*) and selected the target site of inactivation based on the known physiology of neurons (*Ohayon et al., 2012*). (4) We used a behavioral paradigm that was naturalistic, yet allowed for quantification of perceptual thresholds. Thus we could provide first causal evidence for the role of ML in real-life face detection. This is despite only unilaterally inactivating 1 out of 12 nodes (*Tsao et al., 2006, 2008*) and allowing free viewing. Receptive fields in posterior IT, the region in which ML is located, are spatially specific (*Kobatake and Tanaka, 1994*), and thus the two hemispheres cover different parts of the visual field (*Rajimehr et al., 2014*). Therefore, unilateral inactivation is expected to affect only part of the visual field (*Afraz et al., 2015*). That significant impairments occur during free viewing emphasizes the causal role of ML in everyday face detection. Effects were specific to faces, with little or no effects on one object category, bodies, that is represented in the direct vicinity of ML (*Popivanov et al., 2012*), and another one, shoes, a newly learned object category with exemplars, as in faces, exhibiting high within-class similarity. Our results thus extend our understanding of ML as a face-specific processing module containing a large number of cells tuned to facial features that are known to be useful for face detection in an important way: highlighting its causal role for real-world face-detection behavior.

## Materials and methods

All procedures conformed to the NIH Guide for Care and Use of Laboratory Animals of the National Institutes of Health. All animals were handled and all procedures and experiments conducted according to approved institutional animal care and use committee (IACUC) protocols at Weill Cornell Medicine (protocol number 2010–0029) and The Rockefeller University (protocol numbers 12585-H and 15849-H). All surgery was performed under isoflurane and a 70% nitrous oxide and 30% oxygen mixture. Every effort was made to minimize suffering.

### Subjects

We studied one male rhesus (Subject O) and one male cynomolgus (Subject I) macaque, 3.5–5 years in age and weighing between 4.5 and 9 kgs. An acrylic cement-based cranial implant (C&B Metabond, Parkell; Palacos LV+G, Zimmer) was anchored to the skull with ceramic screws (Thomas Recording or Rogue Research) using sterile surgical methods under standard anesthetic and postoperative protocols. A custom MRI-compatible headpost made of Ultem (SABIC) was then secured in each implant.

### MRI imaging

We acquired all MRI data on a 3T Siemens Tim Trio MRI scanner and used an AC88 gradient insert (Siemens) for functional scans. A dose of Feraheme (AMAG Pharmaceuticals) containing 8-10 mg/kg of iron was injected into a saphenous vein before each functional scan to increase functional contrast. Functional images were obtained with a custom eight-channel surface coil (Lawrence Wald, MGH Martinos Center) and a gradient-echo echo-planar imaging sequence with 54 horizontal slices, a $96 \times 96$ in-plane matrix, an isotropic resolution of 1 mm$^3$, TR = 2 s, TE = 16 ms, and 2x GRAPPA

acceleration. For anatomical registration and cortical surface reconstruction, we obtained images using a custom single-channel surface coil and a T1-weighted magnetization-prepared rapid gradient echo sequence (MPRAGE) with 240 sagittal slices, an in-plane matrix of 256 × 256, and isotropic resolution of 0.5 mm$^3$ (averaged over four to six scans) under isoflurane anesthesia. After each muscimol injection, we acquired T1-weighted images using a fast MPRAGE sequence with 192 slices, an in-plane matrix of 256 × 256, and isotropic resolution of 0.5 mm$^3$ (one or two repetitions).

## Visual stimuli

Stimuli for fMRI imaging were projected on a screen placed ~35 cm in front of the subject. Stimuli consisted of grayscale pictures of rhesus macaque faces, cynomolgus macaque faces, fruits and vegetables, monkey bodies, manmade objects, and phase-scrambled versions of the fruit and vegetable pictures, presented in separate 24 s blocks. Each block of this set consisted of 15 exemplar images shown for 0.4 s, with each exemplar repeated four times in a block. Images were matched within and across all categories for total screen area, placed on a pink noise background and normalized for luminance and frequency amplitude using the SHINE toolbox (http://www.mapageweb.umontreal.ca/gosselif/shine; *Willenbockel et al., 2010*). The pink noise background subtended 15° visual angle, and the average foreground image subtended ~6° visual angle. A Latin square design was used to create six orders of the stimulus blocks, and the relative position of each block within a run and the identity of the immediately preceding block were balanced.

## fMRI analysis

We used standard, previously published analyses to localize the face patches (*Tsao et al., 2006*). Briefly, we used standard preprocessing methods to correct for artifacts common to imaging head-fixed, awake macaques. We corrected for motion in three dimensions, shifts, rotations, and warping, and aligned the functional images to the high-resolution anatomical images. We used a sign-reversed hemodynamic response function to account for MION increasing contrast by decreasing the local MR signal. A general linear model was then fit, which included predictors for each stimulus type, as well as nuisance predictors for the motion parameters. To map the face patches, we used the contrast [(rhesus faces + cynomolgus faces) > all other object categories]. We constructed a model of each subject's cortical surface from the high-resolution anatomical scans using FreeSurfer (*Fischl, 2012*). The statistical significance values of the differential signal changes from the functional scans were projected onto the high-resolution anatomical volume and surface model for visualization.

## Behavioral training

We trained subjects on a touchscreen-based object detection task. Stimuli were presented using MATLAB software (Kofiko, provided by Dr. Shay Ohayon, MIT; https://github.com/shayo/Kofiko) on an Elo IntelliTouch 1537L 15-inch LCD touchscreen (Tyco Electronics) mounted to the subject's cage. The subject initiated the trial by touching a central disc (4.3 cm dia.), the color of which indicated the target category (red-faces; blue-body parts; green-shoes). On initiating the trial, subjects were presented with a stimulus array consisting of 24 objects, one of which belonged to the target category, presented at 36 possible random locations, balanced to ensure that each quadrant had an equal number of stimuli. The objects were alpha-blended into a cluttered background, based on a set of visual textures (*Portilla and Simoncelli, 2000*) pre-generated using exemplar stimuli from the three target categories. For each trial, a random 600 × 600 region of one of 32 pre-generated textures, at one of four possible rotations, was used as the background. The texture occupied a 26 × 26 cm area of the touchscreen. Subjects were free to choose viewing distance, but typically positioned themselves to minimize movement required to reach the centrally placed reward spout, resulting in a viewing distance of ~20–25 cm. At that viewing distance, the full extent of the stimulus array subtended a visual angle of ~54°–64°. Individual stimuli varied in size and subtended ~4°–7° of visual angle. Subjects had 3 s to choose the correct target object, upon which they were rewarded with a small amount of juice or water. Wrong choices, but not time-outs, resulted in an increased inter-trial delay of 0.5–1.5 s. Sessions were organized into interleaving blocks for each target category (faces, bodies, or shoes), with each block consisting of 10 stimuli at the 10 visibility

levels in random order. Subjects were trained until they performed ~800–1000 trials per 2 hr session, and attained at least 75% face-detection performance at the highest visibility level.

## Behavioral stimuli

Grayscale images of human faces were constructed from a multiracial face database (http://wiki.cnbc.cmu.edu/Face_Place, Michael Tarr). We chose 32 faces with neutral expression, uniform lighting, and viewing angle ranging from half profile to frontal views. Images included hair and in some cases, facial hair. On each face trial, we further applied a random scaling factor (0.5x – 1.5x) and random in-plane rotation (±45 deg.). The goal of these manipulations was to minimize the consistent spatial occurrence of low-level features and ensure that face stimuli possessed highly variable outlines. We used an initial set of 16 frontal face stimuli to train the subjects to criterion, and then switched to this new set just before commencing the inactivation experiments. No performance decrease was triggered by this switch, suggesting that the subjects had learned the stimulus category, and not the individual exemplars. For bodies, we used a set of 16 images consisting of various views of macaque torsos and limbs. Shoe images were downloaded from various online sources. We chose these categories to control for two effects – bodies controlled for the ethological relevance of the stimuli, and shoes controlled for the global shape similarity of objects. The target stimuli did not overlap with the stimuli used for mapping the face patches. Distractor images were drawn from a set of 178 images of airplanes, cars, chairs, flowers, fruits, manmade objects, motorbikes, familiar animal toys, technological objects, Fribbles, and Yadgits (*Barry et al., 2014*). The entire stimulus set was normalized for luminance and frequency amplitude using the SHINE toolbox. All stimuli were alpha-blended into the cluttered background using a transparency mask with feathered edges to achieve smooth blending. We randomized the location of the objects to one of 36 equally spaced locations on the background for every trial, and ensured that: (1) objects did not overlap, and (2) equal numbers of objects were present in each quadrant to maintain a uniform extent and density of visual search.

## Task design considerations

The behavioral task used was the result of theoretical considerations and refinement from pilot behavioral experiments. The visual texture background was introduced to both disrupt the normal coarse contrast relationships in faces, and to introduce face-like coarse contrast relationships at non-target locations. We reasoned that by doing so, pooling over a large number of ML neurons might become necessary for successful task performance. In pilot experiments without a background or with pixel-noise background, animals detected faces rapidly (<1 s) and at high performance levels for all stimulus visibility levels tested. Imposing the texture background resulted in a more typically shaped psychometric curve. Even with a textured background, when a time constraint was not imposed on behavioral response, subjects could perform the face-detection task at near-perfect levels at moderate to high levels of stimulus visibility. However, we noticed that the response time distributions were multi-peaked and had a heavy tail, indicating that the subjects may perform several ~1.5 s-long scans of the array until the target was detected. We imposed a hard time constraint of 3 s, giving subjects time for about two scans of the array. Trials that would have taken longer response times (had the time constraint not been imposed) were thus converted to timeouts or misses attributable to forced guessing, allowing us to obtain a psychometric function that better reflects rapid face detection in scenes.

## Muscimol injection

After functional mapping, MR-compatible chambers were implanted over the functionally identified location of ML under anesthesia, and a craniotomy (~6 mm dia.) was performed within the chamber under sedation just before the inactivation experiments. We injected 5 μL (8 μL in one experiment) of a cocktail of 5 μg/μL muscimol (Sigma Aldrich) and 5 mM gadolinium (Magnevist) in phosphate-buffered saline. Injections were performed through a 32-gauge metal cannula (Hamilton) attached to a precision microinfusion pump (World Precision Instruments) using non-compressible fused silica tubing (Polymicro) and PEEK connectors (Labsmith), adapted from an earlier design (*Noudoost and Moore, 2011*). The cannula was introduced into the cortex via a custom 3D-printed grid, using MAT-LAB-based planning software (*Ohayon and Tsao, 2012*) to select the appropriate grid hole. We

typically injected in 0.2 µL steps, at a rate of 0.01 µL/s, with 20–30 s between steps (average rate of 0.4 µL /min), spread over three cortical depths spaced 500 µm apart (1.6–1.8 µL per depth, deepest first). We waited 3–5 min after each set of injections before moving the cannula to a different depth or withdrawing the cannula from the cortex. The entire procedure lasted ~25 min. Immediately after the injections, the subject was transferred to the MRI scanner to obtain anatomical scans, after which the subject was transferred to its home cage where the touchscreen was already mounted. Subjects thus performed the task between ~1 and 3 hr from the end of the injection procedure. This window was well within the time period of maximal muscimol effects (~0.5 hr to ~4 hr) (*Hikosaka and Wurtz, 1985*; *Lomber, 1999*; *Dias and Segraves, 1999*; *Sommer and Wurtz, 2004*). Off-target sites included locations dorsal to the STS (n = 3; one into STS and two in gray matter 3.0 mm dorsal to STS), deeper locations comprising the white matter beneath ML (n = 3; depths 4–5.5 mm from cortical surface), and locations posterior to ML (n = 4; 2.9–3.6 mm from the center of ML at t > 20). The first sub-group of off-target injections was intended to function as sham injections, as the animal undergoes identical experimental procedures without inactivation of visual areas. We used off-target shams rather than injecting saline into ML to minimize risks of causing tissue damage and to preserve the face patches for future electrophysiological studies. We performed 11 on-target injections and 10 off-target injections in three hemispheres. To evaluate overlap between the muscimol injection and the functionally defined bounds of ML, we used AFNI to construct a binary mask volume that isolated t-values greater than 20 in the [(rhesus faces + cynomolgus faces) > all other objects] contrast. We resampled this mask volume to the resolution of the anatomical volume, and manually selected voxels corresponding to ML. In the anatomical volume, we first constructed a manual mask that included the area around the injection site. Then, we determined the maximum value of Gd brightness within this mask, and constructed a second binary mask that included all values greater than 50% of the maximum value (full-width at half maximum of the injection site). We then used AFNI to compute the volume of the injection (in voxels), and the volumetric overlap between the injection and ML (at t > 20). The injection overlap was defined as the volume of overlap divided by the volume of the injection. ML overlap was defined as the volume of overlap divided by the volume of ML.

## Behavior and behavioral data analysis

Subjects were required to select, within 3 s, an object belonging to the target category from an array of 24 objects, blended into a cluttered background at one of 10 visibility levels (*Figure 1A,B*). Subjects had been trained on three target categories: human faces, macaque bodies, or shoes. Each trial had three possible outcomes – target object selected, distractor object selected, or a timeout (3s elapsed without a touch). For all analyses, we focused on the proportion of correct trials. We first compared the average performance difference between each experimental condition (inactivated, recovery or off-target) and the paired baseline condition for each category, using FDR-corrected paired t-tests to evaluate statistical significance. We then fit psychometric functions to the data from individual sessions in each experimental condition for each stimulus category. These functions were of the form (*Wichmann and Hill, 2001*):

$$\psi(x; \alpha, \beta, \lambda) = \gamma + (1 - \gamma - \lambda) * F(x; \alpha, \beta) \tag{1}$$

where F is the Weibull function, defined as $F(x; \alpha, \beta) = 1 - \exp\left(-\left(\frac{x}{\alpha}\right)^{\beta}\right)$, $\alpha$ is the shift parameter, $\beta$ is the slope parameter, $\lambda$ is the lapse rate, and $\gamma$ = 1/24 = 0.0417 is the chance performance level (assuming that the animal initiates trials with the intention of finishing them). A logit transform was applied to all proportional data (such as percent correct) before applying parametric statistical tests. Paired t-tests were used to evaluate whether fit parameters were different for each experimental condition from baseline. Thresholds were derived from these psychometric fits. We used ANOVAs with post-hoc Tukey's tests to compare performance deficits across experimental conditions and stimulus categories. In preliminary analyses, we noticed that performance deficits decreased as we exposed the subject to an increasing number of inactivation sessions. To control for this effect, we also performed an ANCOVA to compare effect sizes between the various groups after accounting for session number, which was modeled as a covariate. All analyses were performed in MATLAB (MathWorks, Natick, MA). We derived equivalent inactivated visibilities from the fit curves, and plotted example face stimuli blended into the background texture at the resulting alpha levels. We

caution that these example stimuli render differently in print and on different monitor types (LCD or CRT).

## Acknowledgements

We thank Shay Ohayon (Massachusetts Institute of Technology) for providing the behavioral control software environment, and the Planner software package for stereotactic targeting. We thank Akinori Ebihara, Rizwan Huq, Sofia Landi, Clark Fisher, Caspar Schwiedrzik, Stephen Shepherd, Pablo Polosecki, Ilaria Sani, Julia Sliwa and Margaret Fabiszak (The Rockefeller University) for assistance with imaging and training; Skye Rasmussen, Leslie Diaz, Alejandra Gonzales, Susan Hinklein, Jorge Fajardo, and Vadim Sherman (The Rockefeller University) for veterinary and technical support, Jon Dyke (Weill Cornell Medicine) for scanner support, and Lawrence L Wald for MRI coil design and support. We thank Melanie Wilke and Igor Kagan for advice on targeted muscimol injections, and Gaby Maimon and Stephen Shepherd for comments on the manuscript. We are grateful for the reviewers' constructive input towards improving this manuscript. This manuscript is based upon work supported by the Center for Brains, Minds and Machines (CBMM), funded by NSF STC award CCF-1231216, funds from The Rockefeller University, and National Eye Institute grant R01 EY021594-01A1 to WAF. WAF is a New York Stem Cell Foundation-Robertson Investigator. SS was supported by a Leon Levy postdoctoral fellowship. WZ was supported by the Pew Fellows Program in Biomedical Sciences and a NIH-International Neuroscience Fellowship F05MH094113.

## Additional information

### Funding

| Funder | Grant reference number | Author |
| --- | --- | --- |
| National Science Foundation | STC award CCF-1231216 | Winrich A Freiwald |
| National Eye Institute | R01 EY021594-01A1 | Winrich A Freiwald |
| New York Stem Cell Foundation | Robertson Investigator | Winrich A Freiwald |
| Pew Charitable Trusts | Fellows and Investigator Programs | Wilbert Zarco Winrich A Freiwald |
| Leon Levy Foundation | Postdoctoral fellowship | Srivatsun Sadagopan |
| National Institutes of Health | International Neuroscience Fellowship F05MH094113 | Wilbert Zarco |

The funders had no role in study design, data collection and interpretation, or the decision to submit the work for publication.

### Author contributions

SS, Conceptualization, Data curation, Software, Formal analysis, Investigation, Visualization, Methodology, Writing—original draft, Writing—review and editing, Designed the experiments, performed functional imaging and inactivation experiments, analyzed data; WZ, Investigation, Methodology, Writing—original draft, Collected behavioral data, performed functional imaging, performed inactivation experiments, assisted with data analysis; WAF, Conceptualization, Supervision, Funding acquisition, Investigation, Project administration, Writing—review and editing, Designed the experiments, performed inactivation experiments, provided inputs for data analysis, wrote and revised the manuscript

### Author ORCIDs

Srivatsun Sadagopan, [iD] http://orcid.org/0000-0002-1116-8728
Wilbert Zarco, [iD] http://orcid.org/0000-0002-3599-0476
Winrich A Freiwald, [iD] http://orcid.org/0000-0001-8456-5030

**Ethics**

Animal experimentation: All procedures conformed to the NIH Guide for Care and Use of Laboratory Animals of the National Institutes of Health. All of the animals were handled and all procedures and experiments conducted according to the approved institutional animal care and use committee (IACUC) protocols at Weill Cornell Medicine (protocol number 2010-0029) and The Rockefeller University (protocol numbers 12585-H and 15849-H). All surgery was performed under isoflurane and in a 70% nitrous oxide and 30% oxygen mixture. Every effort was made to minimize suffering.

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
