## [Decision Letter]

Thank you for submitting your article "A Causal Relationship Between Face-Patch Activity and Face-Detection Behavior" for consideration by *eLife*. Your article has been reviewed by two peer reviewers, and the evaluation has been overseen by a Reviewing Editor (who separately reviewed the paper) and by David Van Essen as the Senior Editor. The following individual involved in review of your submission has agreed to reveal his identity: Rufin Vogels (Reviewer #1).

The reviewers have discussed the reviews with one another and the Reviewing Editor has drafted this decision to help you prepare a revised submission.

Summary:

The present study shows that inactivation of the macaque face selective area ML impairs the detection of faces in cluttered displays. The main finding of the study is that ML plays a causal role in face detection.

The reviewers found the work to be interesting and worthwhile, and potentially publishable in *eLife* after appropriate revision. However, they also had several concerns about methods and interpretation of the results that must be addressed before the paper can be considered for publication.

Essential revisions:

1) Most importantly, one reviewer and the reviewing editor were both concerned about the behavioral procedures used in the experiment, and the apparent lack of head or eye tracking during the experiment. The animals apparently freely viewed the stimulus during the 3 second response period, and no attempt was made to deal with variability in post-processing or analysis. In most vision studies the eyes are tracked closely to ensure that eye movement differences across conditions could not have accounted for the results, but apparently that was not done here. Could this have affected the results? If the eye movement data are available then the paper should be revised appropriately, incorporating those data. If not, then the authors should include a detailed rebuttal indicating why this is not a concern.

2) The behavioral data from individual hemispheres/animals should be shown so that readers can gain a better sense of how consistent the impairment is across hemispheres/animals. The overlap between the estimated injection site and the fMRI defined face patches for each individual hemisphere should also be shown.

3) Although the largest effects seem to occur for faces, inspection of Figure 3 suggests that inactivation affects detection of all three categories, although faces appear most affected. Thus, it is unclear whether the effect is truly limited to faces, or whether it a smaller impairment is also found in other categories. This issue should be addressed clearly in revision.

4) All reviewers had concerns about the location of the muscimol injections in both experimental and control conditions. This information needs to be provided in a very clear form, and the interpretation of the results of each specific injection should be made clear.

*Reviewer #1:*

The present study has several shortcomings, however. Although I feel that these drawbacks do not affect their main result – an impairment of face detection when inactivating ML -, they make a quantitative assessment of the impairment impossible, which is a missed opportunity. First, their "naturalistic" task suffers from a major problem when one wants to quantify detection behavior: if the animal fails to respond to the display that does not mean that the animal did not detect the stimulus. The authors did not measure eye movements of the animals during the detection task and thus an unknown portion of "no go" responses may have resulted not from detection failures but simply from the animals not looking at the display or not performing the task. This causes problems when attempting to compute the proportion of correct trials: what to take as an "executed" trial?

Second, they made unilateral inactivations which will have affected mainly the contralateral visual field (see Afraz et al., 2015). Thus, whether the target stimulus was in the affected part of the visual field will depend on the fixation pattern, which may have varied from trial to trial. This will have added noise to the psychometric functions for the inactivated condition and may have underestimated the deficit to an unknown degree. The data show that ML activation has an effect, but the size of the effect cannot be estimated.

Third, the authors should show the behavioral data of the individual hemispheres/monkeys so that one can appreciate how consistent the impairment is across hemispheres/animals. Furthermore, they should show the overlap between the estimated injection site and the fMRI defined face patches for each individual hemisphere. Related to this, it appears from Figure 1 that in that monkey the injections were rather posterior and at best affected only part of ML. Is this true?

Figure 3 suggest an impairment after each muscimol injection, although the effect is greater for face detection after ML injection than for the other 3 conditions. Could this aspecific part of the effect result from the possibility that the animals did not work well after the injection and following the MRI scan? Control injections with saline – standard in these sort of experiments – could have answered this concern.

I am puzzled by the site of the control injections: dorsal to the STS (where? In gray matter? Non-visual!) and deeper locations in the white matter near ML (subsection “Muscimol injection”). Especially the latter is puzzling since fibers are not affected by muscimol. What was the purpose of this? Sham injections? It would have been more informative to inject in nonface-patch regions of IT.

*Reviewer #2:*

Sadagopan and colleagues studied the behavioral effects that result from inactivation of a face selective area in the temporal lobe of monkeys. The results show that inactivating area ML results in an increased threshold for face detection. Importantly, this increased detection threshold seems to be specific for monkey faces and not other visual categories such as body parts or shoes. In addition, the authors performed muscimol injections in nearby sites that did not affect face detection, demonstrating that the effect is also site specific.

This is a carefully planned experiment that convincingly demonstrates the causal role that area ML has in the processing and detection of visual information related to faces. I have some comments that the authors could address to further improve the manuscript.

1) It is clear that ML inactivation has its strongest effect on the detection of faces. However, Figure 3 shows that ML inactivation also affects the detection of body and shoe categories. These non-selective effects did not reach statistical significance but I think that some readers will find hints of threshold increases for the shoe and body categories. I think no reasonable reader will expect 100% selectivity in any cortical inactivation study. Maybe the authors could make a more guarded interpretation in relation to category selectivity, or maybe they could comment on this.

2) Figure 3. There seems to be no data in relation to performance obtained from off-target inactivation in the shoe and body categories. Why is this?

3) I wonder if the authors have data about the location and spread of the off-target injections. A schematic like the one in Figure 1 but for the off-target inactivation could help readers evaluate how off were the these injections from the ML area.

4) From the data on Figure 2 one can imagine that there were some off-target and recovery sessions that reached a statistical significant effect. Am I correct? Please mark them on those panels.

5) The analyses described in the sixth paragraph of the Results and Discussion seemed redundant to me. Yes, a displacement in detection threshold can be illustrated as a reduction in face visibility but I am not sure what it is gained by doing that.

6) The effects of muscimol decreased with each exposure. This is an interesting finding that I think is worth showing in a data figure. Please show the effect of muscimol as a function of experimental session.

[Editors' note: further revisions were requested prior to acceptance, as described below.]

Thank you for resubmitting your work entitled "A Causal Relationship Between Face-Patch Activity and Face-Detection Behavior" for further consideration at *eLife*. Your revised article has been favorably evaluated by David Van Essen (Senior editor), a Reviewing editor, and two reviewers.

The manuscript has been improved but there are some remaining issues that need to be addressed before a final decision can be made, as outlined below:

1) Please address the injection overlap issue identified by the first reviewer, and if possible provide a more quantitative relationship between inactivation and behavior.

*Reviewer #1:*

Although the authors attempt to defend the use of a "naturalistic" free-viewing task to study the effect of inactivation of a face patch (located in a region with a rough retinotopy), I still feel that this study would have provided more information on the contribution of that face patch to face detection if they had use a well-controlled task (with at least proper eye movement control). Despite this shortcoming, the study shows clear face detection performance deficits, which is a valuable addition to the literature.

I have the following comment on the revised text, which needs to be addressed.

The injection overlap was defined as the volume of overlap (between ML and injection) divided by the injection volume. However, this implies that a small injection volume inside a larger ML volume will get a 100% overlap, while only part of ML was inactivated. They should also provide the proportion of overlap computed relative to the ML volume. This metric reflects how much of ML was inactivated, which is unknown now (especially given the "distorted" map shown in Figure 1), but is important for evaluation of the behavioral effects.

*Reviewer #2:*

The authors have addressed my questions and concerns satisfactorily.

1) I think they now have a more balanced discussion on the selectivity of the muscimol inactivations.

2) They now provide data on the overlap between off-target injections and ML area for each hemisphere.

However, they do not discuss my query: "6) The effects of muscimol decreased with each exposure. This is an interesting finding that I think is worth showing in a data figure. Please show the effect of muscimol as a function of experimentalsession."

I think they missed pasting the response on the Response to Reviewers file because this effect is now shown and analyzed in the revised version of the manuscript (new panel on Figure 3).

[Editors' note: further revisions were requested prior to acceptance, as described below.]

Thank you for resubmitting your work entitled "A Causal Relationship Between Face-Patch Activity and Face-Detection Behavior" for further consideration at *eLife*. Your revised article has been favorably evaluated by David Van Essen (Senior Editor), a Reviewing Editor, and two reviewers.

The reviewers were very happy with your revisions and we are ready to accept the manuscript. However, one reviewer requested a small technical change to correct some missing information. In the manuscript you currently report one degree of freedom for the ANCOVA, but there should be two degrees of freedom reported [e.g., in the following format F(df1,df2) = x]. Please provide the other degree of freedom here, or provide an explanation for why this is not required.

After we receive your revised submission we will be happy to publish this work.

---

## [Author Response]

*Essential revisions:*

*1) Most importantly, one reviewer and the reviewing editor were both concerned about the behavioral procedures used in the experiment, and the apparent lack of head or eye tracking during the experiment. The animals apparently freely viewed the stimulus during the 3 second response period, and no attempt was made to deal with variability in post-processing or analysis. In most vision studies the eyes are tracked closely to ensure that eye movement differences across conditions could not have accounted for the results, but apparently that was not done here. Could this have affected the results? If the eye movement data are available then the paper should be revised appropriately, incorporating those data. If not, then the authors should include a detailed rebuttal indicating why this is not a concern.*

We will address the general and most important concern (eye-tracking) raised here in quite some detail. In brief, our study is a behavioral visual search study. Much of the classical and recent behavioral work on visual search (in humans and monkeys) has been conducted without eye movement monitoring. The analyses of behavior we are using here follows these established techniques. Like the reviewer, we do not think that lack of eye tracking detracts from the main results of the study.

In our lab, we use eye tracking regularly, since it is necessary for most of the paradigms we employ. This is especially the case for electrophysiological recordings, to account for variability of neural firing rates arising from shifts of receptive fields over the stimulus array.

But because the study in this manuscript is purely behavioral, we could make a fundamental design choice when we started this study, given that reliable eye tracking for freely moving primates is not available: 1) do a traditional head-fixed experiment with eye-tracking, or 2) exploit a natural visual search behavior by using a touchscreen. We deliberately chose the latter option, fully recognizing that we were trading off the advantages we gained (almost unconstrained natural behavior) against disadvantages (lack of eye tracking).

We did so mainly for one reason: in the landscape of the existing literature on face processing, which virtually exclusively relies on the head-fixed preparation, we believed we could be truly innovative and contribute to the ongoing debates about the relevance of face-representations for real-world face- recognition behavior by studying the dependence of a natural behavior on neural activity. We believe that this is a transformative innovation for the field of face recognition as a whole, and of fundamental value to the current study.

As we mentioned earlier, this choice was influenced by many classical and recent studies on visual search and object recognition. Many recent human behavioral studies on visual search and object recognition are conducted on engines such as Amazon MTurk where there is no control on gaze, experimental settings, or distractions. For example, a recent study on core object recognition obtained human performance data from MTurk by pooling the responses of a large number (~600) of human subjects performing a small number (~100) of trials, without eye tracking data (Rajalingham et al., J. Neurosci. 2015). In the same study, they performed monkey object recognition experiments on a touchscreen (one subject) or with eye tracking (one subject), and found that performance was similar between the monkey subjects. Indeed, the study concluded using these data that core object recognition was similar between humans and monkeys. This illustrates that studies without eye tracking can be consistent with those using eye tracking and highly informative. A possible reason for this high degree of similarity is that humans and monkeys naturally tend to foveate the most interesting or task-relevant regions of the visual field.

As we will detail below in our response to specific reviewer comments, we agree with the reviewer that this means that the “true” underlying effect size is likely larger than what we report. Thus all our behavioral quantifications are more precisely seen as lower bound estimates. But the true value of an inactivation study like ours really lies in providing evidence for the existence and for the specificity of an effect during a naturalistic behavior, and this is what the current study accomplishes.

We further argue that the difficulties quantifying behavior that reviewer 1 points out are difficulties resulting from the logic of go/no-go tasks, not of the lack of eye position measurements. Recognizing the need for tight behavioral control in the absence of eye position monitoring, we designed the task to minimize potential for unwanted distractions by: 1) Requiring the animals to voluntarily initiate trials; 2) Requiring subjects to direct their gaze to the center of the screen and perform a deliberate and accurate movement to initiate the trial; 3) Requiring subjects to pay attention to the color of the central cue to determine target identity; 4) Requiring subjects to complete the task in a short time span; and 5) Randomizing visibility levels within blocks and interleaving block types to prevent systematic biases. Through post-hoc analysis we determined that: 1) The animal performed similar numbers of trials on control and inactivation days suggesting no change in motivation level; 2) Animals performed at a high rate on high-visibility trials on all days, suggesting no distraction at the time scale of trials. The experimental configuration ensured that the screen covered a large portion of the animal’s visual field, minimizing space available for distractions. Therefore, we believe that we have minimized the effect of extraneous factors on behavior.

From the information we had provided with the first submission, it was probably difficult for the reviewers to get a sense of the behavioral paradigm and the degree of focus subjects had to pay in order to succeed at it. Perhaps the advantages of the touchscreen were overstated in the original submission; we have now revised the manuscript for a more balanced view. Overall, we believe that we are in agreement with the reviewer that this does not detract from the main results of our study.

Please see our responses to reviewer 1 for details. We have also made the following changes in the revised manuscript:

“The free-viewing nature of the task exploits natural visual search behavior to demonstrate the causal dependence of this behavior on face patch activity. […] The experimental configuration ensured that the screen covered a large portion of the animal’s visual field, minimizing space available for distractions. Therefore, we believe that we have minimized the effects of extraneous sources on behavior.”

*2) The behavioral data from individual hemispheres/animals should be shown so that readers can gain a better sense of how consistent the impairment is across hemispheres/animals. The overlap between the estimated injection site and the fMRI defined face patches for each individual hemisphere should also be shown.*

A) We have now included two new figure panels (Figure 2) that now show the data from individual hemispheres/animals, demonstrating the consistency of the effect of inactivating ML on face detection.

“In Figure 2, we plot the same quantities as above separated into each hemisphere and animal to demonstrate that the reduction in performance was consistent between subjects and inactivated hemispheres. At the single-hemisphere level, we observed significant differences mainly at intermediate visibility levels (p<0.05; FDR-corrected paired t-test after logit transform).”

We have now quantified the overlap between the injection site and ML, and provided a summary in new figure panel Figure 1 that is also broken down into individual hemispheres/animals.

“On average, 76% of the half-maximal extent of the injection volume overlapped with ML volume exceeding a threshold of t>20.”

“The off-target injection volumes barely overlapped with ML (9.6%; Figure 1) or with any other face patch, and consequently, we observed only a weak and insignificant reduction in detection performance (Figure 2 and Figure 3; blue).”

“To evaluate overlap between the muscimol injection and the functionally-defined bounds of ML, we used AFNI to construct a binary mask volume that isolated t-values greater than 20 in the (rhesus faces + cynomolgus faces) > all other objects contrast. […] The injection overlap was then defined as the volume of overlap divided by the volume of the injection.”

For statistical power, we now correct for multiple comparisons using the false discovery rate (FDR) instead of using the Bonferroni-Holm correction (mentioned in manuscript wherever changed).

*3) Although the largest effects seem to occur for faces, inspection of Figure 3 suggests that inactivation affects detection of all three categories, although faces appear most affected. Thus, it is unclear whether the effect is truly limited to faces, or whether it a smaller impairment is also found in other categories. This issue should be addressed clearly in revision.*

We approached this question in three ways:

A) We grouped the ‘body’ and ‘shoe’ categories into a single ‘non-face’ category and used the same ANCOVA analysis that yielded significant results for the Face/body/shoe analysis to test whether there was a significant difference between non-face control and inactivated conditions. We did not find a significant difference. A similarly grouped ANOVA analysis of thresholds also failed to reveal any non- face effects if inactivation.

“Although ML inactivation did not cause statistically significant performance deficits or threshold shifts for the other object categories, it is possible that the data might contain a non-specific component, as indicated by the non-zero values of performance deficits and thresholds for body parts and shoes (Figure 3). […] Similarly, grouping the ‘body’ and ‘shoe’ categories in the ANOVA analysis of threshold failed to yield a significant difference in threshold.”

B) To determine if the cause was a change in motivation on inactivation days (per reviewer 1), we compared the number of trials the animal initiated on control and inactivation days, and did not find a significant difference.

“If a weak non-specific effect truly existed, we could not attribute this to the motivation level of the animal post injection and MRI scan, as there were no significant differences between the total number of trials the animal initiated on control or inactivation days (paired t-test, p = 0.47).”

C) We have tempered our claim with regard to category selectivity (per reviewer 2) by discussing a potential drawback of our method:

“Alternately, although 76% of the injection volume was contained within ML, it is possible that some muscimol diffused out of ML and into neighboring body patches (Popivanov et al., 2012) in some cases, causing a smaller, non-significant effect on that object category. “

*4) All reviewers had concerns about the location of the muscimol injections in both experimental and control conditions. This information needs to be provided in a very clear form, and the interpretation of the results of each specific injection should be made clear.*

Please see response to Essential revisions #1. We have now quantified the overlap between the injection and ML in each case, and present these data for individual subjects/hemispheres in new figure panel Figure 1. As we demonstrate, the experimental muscimol injections largely overlapped with ML (76% overlap) whereas the control injections did not (9.6% overlap).

Secondly, we now describe the locations of the control injections in much more detail. We chose these off-target sites because we attempted to generate a range of injections that were in the near vicinity of the face patch. The sham injections (n=10) fell into three broad subgroups: injections into the white matter directly beneath the face patch (n=3), injections dorsal to the STS (n=3; 2 of which were in gray matter), and injections posterior and lateral to face patch ML (n=4). The first subgroup was intended to function as a sham group, where the animal experiences identical experimental procedures, has muscimol in the brain, but likely does not affect visual cortex activity. In the second and third subgroup, non-face selective areas were inactivated. We did not find statistically significant differences between these subgroups, which is why they were combined into an ‘off-target’ group for analysis. Please see our response to reviewer 1 for more details.

The revised manuscript now contains these details:

“Off-target sites included locations dorsal to the STS (n = 3, 2 in gray matter), deeper locations comprising the white matter beneath ML (n = 3), and locations posterior to ML (n = 4). […] We used off-target shams rather than injecting saline into ML in order to minimize risks of causing tissue damage and preserve the face patches for future electrophysiological studies.”

*Reviewer #1:*

*The present study has several shortcomings, however. Although I feel that these drawbacks do not affect their main result – an impairment of face detection when inactivating ML -, they make a quantitative assessment of the impairment impossible, which is a missed opportunity. First, their "naturalistic" task suffers from a major problem when one wants to quantify detection behavior: if the animal fails to respond to the display that does not mean that the animal did not detect the stimulus. The authors did not measure eye movements of the animals during the detection task and thus an unknown portion of "no go" responses may have resulted not from detection failures but simply from the animals not looking at the display or not performing the task. This causes problems when attempting to compute the proportion of correct trials: what to take as an "executed" trial?*

We agree that the lack of eye-tracking in our data was not sufficiently discussed in the original submission. As we stated earlier in the “Essential revisions” section, this is a question that we have asked ourselves since the inception of this project. In the majority of paradigms we use in the lab, eye- tracking plays a central role. However, we realized that the behavioral study we planned had a lot of similarity to human visual search and object recognition studies that are based on free viewing, and realized that by using a “naturalistic” task, we could add a lot more to the debate on the dependence of behavior on neural activity than would have been possible using head-fixed preparations.

We emphasize the reason we decided to use a touchscreen-based, free-viewing task: previously published studies that have manipulated the activity of ventral stream areas have almost exclusively used a head-fixed preparation. As the reviewer points out in his comment below, these studies (such as the ones by Afraz) have shed light on specific areas of the visual field that are impacted by inactivating the topographically corresponding areas of cortex. Our approach is complementary to the head-fixed studies which may offer better quantification of behavior, but at the expense of a constrained experimental prep.

Having chosen a free-viewing paradigm, we designed our task to minimize possibilities of distraction. First, animals initiated individual trials voluntarily by touching a central disc that was about the same size as the target objects. This required that the animal direct its attention and gaze to the center of the screen, and perform a deliberate and precise action to initiate a trial. The color of the central disc cued the animal to the target category; therefore, it was beneficial for the animal to pay attention to the disc while initiating the trial. For these reasons, our design ensures that the animal is ready and attentive when initiating the trial. From this, we conclude that the animal voluntarily initiates trials with the intent to complete them.

As with most behavioral studies, we worked under the assumption that the animal sought to maximize its reward within the time available for behavior. In our case, the animals knew that they had ~2 hours to complete the task (trained as such because of the time scale of muscimol effectiveness). On a day to day basis, we did not find significant differences in the number of trials completed by the animal in the control and inactivation days, suggesting that motivation on these two days was roughly constant. On the timescale of blocks, the animal still performed at 75% for high visibility levels, and performance decreased <10% after inactivation. Since the visibilities are interleaved and randomized, it is unlikely that distractions were happening at this time scale as well.

Thus, the only remaining theoretical possibility is that the animal was distracted after it initiated a trial (with intention to complete). Our task design minimizes this possibility because: 1) the animal only had 3 seconds to respond, restricting the time available for distractions, and 2) the touchscreen covers a large portion of the visual field, restricting the space available for distractions.

Finally, what the reviewer is pointing out here is a general problem with go/no-go tasks that is logically difficult to address even with eye tracking. When an animal (or human) does not generate a ‘go’ response after voluntarily initiating a trial, is it the result of a true rejection, or because of other factors such as a distraction that shifts the covert attention of the animal?

Therefore, while we agree with the reviewer that eye-tracking could offer better quantification of behavior, we believe that: 1) given previous studies in head-fixed animals, we could contribute innovatively by using free-viewing behavior, and 2) we designed our task to minimize potential for distractions. We also agree that this should not affect our main result.

In the revised manuscript, we now discuss this issue at some length:

“The free-viewing nature of the task exploits natural visual search behavior to demonstrate the causal dependence of this behavior on face patch activity. […] Therefore, while the lack of eye tracking does not affect the main results of our study, it introduces two sources of variability into our data.”

“Second, the lack of eye-tracking limits our ability to quantify behavioral strategies that may have been used by the subject to solve the task. […] Therefore, we believe that we have minimized the effects of extraneous sources on behavior.”

*Second, they made unilateral inactivations which will have affected mainly the contralateral visual field (see Afraz et al., 2015). Thus, whether the target stimulus was in the affected part of the visual field will depend on the fixation pattern, which may have varied from trial to trial. This will have added noise to the psychometric functions for the inactivated condition and may have underestimated the deficit to an unknown degree. The data show that ML activation has an effect, but the size of the effect cannot be estimated.*

The reviewer is correct. Our goal was to provide a lower bound on the effect of inactivation during natural behavior, not to estimate the effect precisely. We believe that we are in agreement with the reviewer in that this does not affect the main conclusions of the study.

We designed our task to test the dependence of one of the most fundamental of behaviors (face detection) on neural activity in a naturalistic context. A few previous studies (for example, Afraz et al., 2015) have demonstrated spatially restricted deficits using head-fixed preparations. As these studies show, and as the reviewer suggests, it is quite likely the case that the deficit is stronger in the contralateral visual hemifield. Nonetheless, the cited study reports drops in behavioral performance in the order of 2.0% to 5.5%, which can be considered modest effect sizes for a head fixed controlled experiment. With our current approach, we have shown that the effect size is *at least* an 8% threshold shift even in conditions where the animal is free to choose an optimal strategy. We think that this is a significant contribution to the debates around the causal effect of neural activity on behavior.

ML is also known to be largely foveal (Janssens et al., 2014, as well as electrophysiological data from our own lab). During visual search, animals and humans foveate a series of points that contain the most task-relevant information. Thus, by affecting ML activity, it is likely that we have affected a critically informative region of the visual field that leads to the observed deficits.

In the original submission, we had perhaps overstated the advantages of the touchscreen. In the revised manuscript, we now present a more balanced discussion of its disadvantages as well:

“The free-viewing nature of the task exploits natural visual search behavior to demonstrate the causal dependence of this behavior on face patch activity. […] Thus, performance deficits may have been higher at target locations that correspond to retinotopic location of the injection, and effect sizes reported here are to be taken as lower bound estimates.”

*Third, the authors should show the behavioral data of the individual hemispheres/monkeys so that one can appreciate how consistent the impairment is across hemispheres/animals. Furthermore, they should show the overlap between the estimated injection site and the fMRI defined face patches for each individual hemisphere. Related to this, it appears from Figure 1 that in that monkey the injections were rather posterior and at best affected only part of ML. Is this true?*

A) We have now included two new figure panels (Figure 2) that now show the data from individual hemispheres/animals, demonstrating the consistency of the effect of inactivating ML on face detection.

“In Figure 2, we plot the same quantities as above separated into each hemisphere and animal to demonstrate that the reduction in performance was consistent between subjects and inactivated hemispheres. At the single-hemisphere level, we observed significant differences mainly at intermediate visibility levels (p<0.05; FDR-corrected paired t-test after logit transform).”

B) We have now quantified the overlap between the injection site and ML, and provided a summary in new figure panel Figure 1 that is also broken down into individual hemispheres/animals.

“On average, 76% of the half-maximal extent of the injection volume overlapped with ML volume exceeding a threshold of t>20.”

“The off-target injection volumes barely overlapped with ML (9.6%; Figure 1) or with any other face patch, and consequently, we observed only a weak and insignificant reduction in detection performance (Figure 2 and Figure 3; blue).”

“To evaluate overlap between the muscimol injection and the functionally- defined bounds of ML, we used AFNI to construct a binary mask volume that isolated t-values greater than 20 in the (rhesus faces + cynomolgus faces) > all other objects contrast. […] We then used AFNI to compute the volume of the injection (in voxels), and the volumetric overlap between the injection and ML (at t>20). The injection overlap was then defined as the volume of overlap divided by the volume of the injection.”

With regard to the overlap of injection sites and ML in Figure 1 – the volumetric representation (Figure 1) is a more realistic depiction of the injection site and ML than the surface visualization, for two reasons. First, the surface transform creates distortions especially at the lips of sulci. Second, we choose a depth mapping for projection (either a specific depth such as halfway in the gray matter, or a maximum projection) that does not present all the information on the display. Experimentally, we used a grid hole that gave us the best access to the center of the ML volume. This particular hemisphere was no different, and as our analysis shows, the injection was contained within ML in all animals.

*Figure 3 suggest an impairment after each muscimol injection, although the effect is greater for face detection after ML injection than for the other 3 conditions. Could this aspecific part of the effect result from the possibility that the animals did not work well after the injection and following the MRI scan? Control injections with saline – standard in these sort of experiments – could have answered this concern.*

We agree that this warrants additional analyses.

To answer the specific concern about motivation level, we compared the total number of trials the animal initiated on control and inactivation days. Animals performed about 900 trials on both days and we did not find a significant difference.

“If a weak non-specific effect truly existed, we could not attribute this to the motivation level of the animal post injection and MRI scan, as there were no significant differences between the total number of trials the animal initiated on control or inactivation days (paired t-test, p = 0.47).”

We did not perform saline injections into ML to minimize the risk of causing any tissue damage within the face patch and to preserve the face patch for future electrophysiological experiments. The off-target white-matter injections were intended to function as sham injections (see below).

Moreover, we conducted additional analyses where we grouped the ‘body’ and ‘shoe’ categories into a single ‘non-face’ category and used the same ANCOVA analysis that yielded significant results for the Face/body/shoe analysis to test whether there was a significant difference between non-face control and inactivated conditions. We did not find a significant difference. A similarly grouped ANOVA analysis of thresholds also failed to reveal any significant effects of inactivation for non-face stimuli.

“Although ML inactivation did not cause statistically significant performance deficits or threshold shifts for the other object categories, it is possible that the data might contain a non-specific component, as indicated by the non-zero values of performance deficits and thresholds for body parts and shoes (Figure 3). […] Similarly, grouping the ‘body’ and ‘shoe’ categories in the ANOVA analysis of threshold failed to yield a significant difference in threshold.”

Therefore, while the effect sizes are indeed non-zero as the reviewer observes, we are not sure if we can draw any conclusions from those data in the absence of statistically significant results. Therefore, we have revised the manuscript with the above additional analyses to help the reader evaluate the data themselves.

Finally, as we demonstrate through new analyses, 76% of the injection volume was contained within ML, and it is possible that some muscimol diffused into neighboring body patches, causing a smaller effect. We now acknowledge this in the revised manuscript.

“Alternately, although 76% of the injection volume was contained within ML, it is possible that some muscimol diffused out of ML and into neighboring body patches (Popivanov et al., 2012) in some cases, causing a smaller, non-significant effect on that object category.”

*I am puzzled by the site of the control injections: dorsal to the STS (where? In gray matter? Non-visual!) and deeper locations in the white matter near ML (subsection “Muscimol injection”). Especially the latter is puzzling since fibers are not affected by muscimol. What was the purpose of this? Sham injections? It would have been more informative to inject in nonface-patch regions of IT.*

We chose these off-target sites because we attempted to generate a range of injections that were in the near vicinity of the face patch. The reviewer is correct in pointing out that these injections were intended to serve both as shams and as controls for inactivating non-face regions of IT cortex, and we now state so clearly in the revised version of the manuscript.

As we outlined earlier, we did not inject saline into the face patch as a control because we wanted to minimize the risk of causing tissue damage within the face patch. We took this precaution because: 1) the volumes of the injections were large (5uL), and 2) the sham and inactivation sessions were interleaved, and we did not want to adversely affect future inactivation sessions or electrophysiological recordings.

The off-target injections (n=10) fell into three broad categories: injections into the white matter directly beneath the face patch (n=3), injections dorsal to the STS (n=3; 2 of which were in gray matter), and injections posterior and lateral to face patch ML (n=4).

As the reviewer rightly points out, the white matter injections were intended to function as shams, as fibers do not express the GABA receptor and should therefore be unaffected by muscimol. These injections replicate all experimental procedures but leaves face patch activity intact. Injections dorsal to the STS again replicate all experimental procedures and do inactivate portions of cortex at the same A- P coordinates as ML, but do not affect face patch activity. Finally, injections posterior and lateral to ML were intended to inactivate non-face but visual regions of IT cortex. We did not find significant differences in the performance deficit between these three groups (ANOVA followed by pairwise post- hoc tests), which is why they were all combined into a single off-target group for analysis. In our new analysis of Injection overlap, the injections dorsal to the STS showed significantly lower overlap with ML (0%) than the other two sham groups.

In the revised manuscript, we now provide the following additional details:

“Off-target sites included locations dorsal to the STS (n = 3, 2 in gray matter), deeper locations comprising the white matter beneath ML (n = 3), and locations posterior to ML (n = 4). […] We used off-target shams rather than injecting saline into ML in order to minimize risks of causing tissue damage and preserve the face patches for future electrophysiological studies.”

*Reviewer #2:*

*[…] 1) It is clear that ML inactivation has its strongest effect on the detection of faces. However, Figure 3 shows that ML inactivation also affects the detection of body and shoe categories. These non-selective effects did not reach statistical significance but I think that some readers will find hints of threshold increases for the shoe and body categories. I think no reasonable reader will expect 100% selectivity in any cortical inactivation study. Maybe the authors could make a more guarded interpretation in relation to category selectivity, or maybe they could comment on this.*

We performed additional analyses to address this observation that the effect sizes are non-zero for body and shoe detection.

First, we grouped the ‘body’ and ‘shoe’ categories into a single ‘non-face’ category to increase statistical power – the number of trials in the non-face category doubles, providing a better estimate of the mean effect size, and the number of categories decrease, making multiple comparisons correction more powerful. We then used the same ANCOVA analysis that yielded significant results for the Face/body/shoe performance deficit analysis to test whether there was a significant difference between non-face control and inactivated conditions. We did not find a significant difference. A similarly grouped ANOVA analysis of thresholds also failed to reveal any non-face effects if inactivation.

“Although ML inactivation did not cause statistically significant performance deficits or threshold shifts for the other object categories, it is possible that the data might contain a non-specific component, as indicated by the non-zero values of performance deficits and thresholds for body parts and shoes (Figure 3). […] Similarly, grouping the ‘body’ and ‘shoe’ categories in the ANOVA analysis of threshold failed to yield a significant difference in threshold.”

Second, to determine if the cause was a change in motivation on inactivation days (per Reviewer 1) that could result in a non-selective decrease in performance, we compared the number of trials the animal initiated on control and inactivation days, and did not find a significant difference.

“If a weak non-specific effect truly existed, we could not attribute this to the motivation level of the animal post injection and MRI scan, as there were no significant differences between the total number of trials the animal initiated on control or inactivation days (paired t-test, p = 0.47).”

Finally, following the reviewer’s suggestion, we offer a more cautious claim with regard to category selectivity by discussing a potential drawback of our method:

“Alternately, although 76% of the injection volume was contained within ML, it is possible that some muscimol diffused out of ML and into neighboring body patches (Popivanov et al., 2012) in some cases, causing a smaller, non-significant effect on that object category.”

“Thus ML inactivation is largely category-selective for faces, and anatomically specific for ML compared to its immediate surrounding.”

*2) Figure 3. There seems to be no data in relation to performance obtained from off-target inactivation in the shoe and body categories. Why is this?*

This is because ‘off-target’ is well-defined for face patch ML as this is functionally mapped, but:

A) is not defined in this study with respect to ‘body’ and ‘shoe’ processing, and

B) ‘off-target’ ML inactivation may not be the same as an ‘off-target’ body injection or ‘off-target’ shoe injection.

Thus, for the purposes of this study and to avoid confusion, we only present analysis of off-target injections with respect to ML.

*3) I wonder if the authors have data about the location and spread of the off-target injections. A schematic like the one in Figure 1 but for the off-target inactivation could help readers evaluate how off were the these injections from the ML area.*

It might be misleading to display off-target data using a similar schematic because the sham injections are at different depths (including some in white matter). The surface mapping procedure typically maps values at a particular depth to the surface image. If we use a ‘max’ mapping, it will appear as though some sham injections are located within the face patch, but only because the difference in depth is ignored. Second, it is difficult to quantify the overlap between the injection and ML because of how the surface is generated from the volume (using triangular tessellation). For these reasons, we used analysis based on the anatomical volumes themselves to determine the bounds of the injection, of face patch ML, and quantify the overlap. Through this analysis, we show that ~76% of ML injection volumes are contained within ML, whereas only an ~10% volume of off-target injections overlap with ML.

We have now quantified the overlap between the injection site and ML, and provided a summary in new figure panel Figure 1 that is also broken down into individual hemispheres/animals.

“On average, 76% of the half-maximal extent of the injection volume overlapped with ML volume exceeding a threshold of t>20.”

“The off-target injection volumes barely overlapped with ML (9.6%; Figure 1) or with any other face patch, and consequently, we observed only a weak and insignificant reduction in detection performance (Figure 2 and Figure 3; blue).”

“To evaluate overlap between the muscimol injection and the functionally- defined bounds of ML, we used AFNI to construct a binary mask volume that isolated t-values greater than 20 in the (rhesus faces + cynomolgus faces) > all other objects contrast. […] The injection overlap was then defined as the volume of overlap divided by the volume of the injection.”

The locations of off-target injections are now better described in the revised manuscript.

“Off-target sites included locations dorsal to the STS (n = 3, 2 in gray matter), deeper locations comprising the white matter beneath ML (n = 3), and locations posterior to ML (n = 4). […] We used off-target shams rather than injecting saline into ML in order to minimize risks of causing tissue damage and preserve the face patches for future electrophysiological studies.”

*4) From the data on Figure 2 one can imagine that there were some off-target and recovery sessions that reached a statistical significant effect. Am I correct? Please mark them on those panels.*

There are some outliers for off-target and recovery sessions in Figure 2 as the reviewer rightly notes. But these represent single sessions, and we are not sure how we can test for statistical significance for these single samples. We analyzed the population of n=10 off-target sessions and n=11 recovery sessions, and found no statistical differences. We are happy to perform any specific analyses the reviewer might have in mind.

*5) The analyses described in the sixth paragraph of the Results and Discussion seemed redundant to me. Yes, a displacement in detection threshold can be illustrated as a reduction in face visibility but I am not sure what it is gained by doing that.*

This analysis was included because to a general reader, it may be unclear what a “11% deficit” or an “8% threshold shift” might represent. By performing these analyses, we transformed these “percent deficits” into actual images, illustrating the perceptually dramatic effects that an 8% threshold shift represents. A ‘small’ perceived effect size (8%) is a perceptually dramatic effect because of the nonlinearity of the psychophysical contrast-response function, and this is illustrated by these analyses.

As both reviewers point out, there was one result that should be completely expected by the displacement in threshold, and we have now stated this explicitly:

“As expected from the increased detection threshold, equivalent inactivated visibilities for faces were significantly lower than baseline visibilities (p<0.01; paired t-test).”

*6) The effects of muscimol decreased with each exposure. This is an interesting finding that I think is worth showing in a data figure. Please show the effect of muscimol as a function of experimental session.*

[Editors' note: further revisions were requested prior to acceptance, as described below.]

*The manuscript has been improved but there are some remaining issues that need to be addressed before a final decision can be made, as outlined below:*

*1) Please address the injection overlap issue identified by the first reviewer, and if possible provide a more quantitative relationship between inactivation and behavior.*

We thank the reviewer for this suggestion. In our past revision, we chose a metric with respect to the injection volume because we had interpreted the concerns of both reviewers in the previous round of reviews as being about the extent to which non-target injections may have leaked into ML. We agree with reviewer 1’s point, however, and for this revision, we:

1) calculated the overlap metric with respect to ML volume inactivated as well,

2) updated Figure 1 to include both overlap metrics, and

3) presented the actual volumes of ML and the injection (at the chosen thresholds) to address the concern that our injections were much smaller in volume than ML.

“The goal of injections was to inactivate as much of ML as possible with as little inactivation of outside regions as possible – and the converse for the off-target injections. […] The corresponding numbers for off-target injections were 4% ML coverage with 10% of the injection contained within ML, and it is thus unlikely that off-target injections, though spaced closely enough to ML to generate an overlap, silenced many face-selective neurons.”

We thank the reviewer for suggesting further analyses of the relationship between the precise properties of the injections and behavioral effects. We performed a number of such analyses. In particular, we correlated the average deficit in face-detection against injection overlap using a couple of different methods. We saw trends, but no significant correlations. In hindsight, this was probably expected, given the relative consistency of injection overlap with ML across experiments, as well as the complexity of injection geometry and of the behavior under study, but it is good to verify this as suggested. The main quantitative difference really is the one between injections inside and outside of area ML. We are now including them in the revised manuscript as follows:

“The mean face detection performance deficit was not significantly correlated with the ML-injection overlap (Spearman’s Rho r_s_ = 0.42; p = 0.1). An ANCOVA analysis with session number as independent groups and ML-injection overlap as a covariate did not reveal a significant effect of ML-injection overlap on the observed deficit (df = 1; F = 1.35; p = 0.25). These data suggest that although there are differences in the extent of ML-injection overlap between sessions, since the central portion of ML was targeted by the injection, performance deficits were largely unaffected by these small differences in injection volume and overlap.”

*Reviewer #1:*

*Although the authors attempt to defend the use of a "naturalistic" free-viewing task to study the effect of inactivation of a face patch (located in a region with a rough retinotopy), I still feel that this study would have provided more information on the contribution of that face patch to face detection if they had use a well-controlled task (with at least proper eye movement control). Despite this shortcoming, the study shows clear face detection performance deficits, which is a valuable addition to the literature.*

I have the following comment on the revised text, which needs to be addressed.

*The injection overlap was defined as the volume of overlap (between ML and injection) divided by the injection volume. However, this implies that a small injection volume inside a larger ML volume will get a 100% overlap, while only part of ML was inactivated. They should also provide the proportion of overlap computed relative to the ML volume. This metric reflects how much of ML was inactivated, which is unknown now (especially given the "distorted" map shown in Figure 1), but is important for evaluation of the behavioral effects.*

We chose the metric relative to injection volume because from the comments of both reviewers in our previous revision, we understood that the concern was about off-target injections diffusing into ML. In this revision, we also provide a metric relative to ML volume, and provide actual volumes of the injection and ML. Figure 1 now plots both metrics for individual hemispheres. We hope that this addresses the reviewers concern about a small injection contained inside a large ML volume.

“The goal of injections was to inactivate as much of ML as possible with as little inactivation of outside regions as possible – and the converse for the off-target injections. […] The corresponding numbers for off-target injections were 4% ML coverage with 10% of the injection contained within ML, and it is thus unlikely that many face-selective neurons were silenced during off-target injections.”

*Reviewer #2: […] However, they do not discuss my query: "6) The effects of muscimol decreased with each exposure. This is an interesting finding that I think is worth showing in a data figure. Please show the effect of muscimol as a function of experimentalsession."*

*I think they missed pasting the response on the Response to Reviewers file because this effect is now shown and analyzed in the revised version of the manuscript (new panel on Figure 3).*

This was indeed the case – we missed pasting our response into the response file. Figure 3 in Revision 1 was meant to address this point.

[Editors' note: further revisions were requested prior to acceptance, as described below.]

*The reviewers were very happy with your revisions and we are ready to accept the manuscript. However, one reviewer requested a small technical change to correct some missing information. In the manuscript you currently report one degree of freedom for the ANCOVA, but there should be two degrees of freedom reported [e.g., in the following format F(df1,df2) = x]. Please provide the other degree of freedom here, or provide an explanation for why this is not required.*

The source of this confusion is now clear to us – while we had taken the reviewer comment from the last review as an analysis problem (running one-way instead of two-way ANOVA), we realize now that it is actually a reporting style issue. We have now updated our reporting to APA guidelines as suggested:

“(F(6, 752)=17.89; p<0.0001)”

“(ANOVA; F(6, 65)=5.74; p<0.0001)”

“face-detection deficit (F(1,102) = 1.35; p = 0.25)”

“(F(1,752)=15.33; p<0.0001)”